# Anti-Doping Knowledge Among Medical Bachelor’s Degree Students in Mexico

**DOI:** 10.3390/healthcare13070742

**Published:** 2025-03-27

**Authors:** Millán Aguilar-Navarro, Alejandro Muñoz, Daniela Rebolledo-Solleiro, Bibiana Moreno-Carranza, Tania Guzman, Javier Díaz-Lara, Arturo Franco-Andrés, Patricia Inda-Icaza, Juan Del Coso

**Affiliations:** 1Doping in Sport Research Group, Exercise and Sports Sciences, Faculty of Health Sciences, Universidad Francisco de Vitoria, 28223 Madrid, Spain; alejandro.munoz@ufv.es (A.M.); arturo.franco@ufv.es (A.F.-A.); 2Escuela Internacional de Medicina, Universidad de Anáhuac, Cancún 77565, Mexico; danielarebolledoso@anahuac.mx; 3Escuela de Ciencias de la Salud, Universidad de Anáhuac, Querétaro 76246, Mexico; bibiana.moreno11@anahuac.mx (B.M.-C.); tania.guzman10@anahuac.mx (T.G.); 4Performance and Sport Rehabilitation Laboratory, Faculty of Sport Sciences, University of Castilla-La Mancha, 45071 Toledo, Spain; javier.diazlara@uclm.es; 5Facultad de Ciencias de la Salud, Universidad Anáhuac, Campus Norte, Mexico City 52786, Mexico; patricia.inda@anahuac.mx; 6Sport Sciences Research Centre, Rey Juan Carlos University, 28943 Fuenlabrada, Spain; juan.delcoso@urjc.es

**Keywords:** sports performance, supporting personnel, doping substances, education

## Abstract

**Background/Objectives**: Universities serve as crucibles for molding future healthcare providers, instilling medical expertise and ethical frameworks crucial for their role as custodians of health. Yet, integrating anti-doping education into university curricula remains largely underexplored, particularly among future physicians. The aim of this research was to evaluate the students’ understanding of anti-doping within the bachelor’s degree in medicine in Mexico. **Methods**: Five hundred and forty-nine bachelor students in medicine (151 males, 351 females, and 7 participants identifying as non-binary) from six universities in Mexico filled out a validated questionnaire regarding general anti-doping knowledge. This questionnaire was an adapted form of the World Anti-Doping Agency’s Play True Quiz and included 36 multiple-choice questions. The results were converted into a scale from 0 to 100 points to evaluate anti-doping knowledge. **Results**: Students scored 55.30 ± 9.08 points (range = 28–83 points). Fourteen questions indicated an error rate higher than 50% within the sample. The course had no impact on the scores achieved in the anti-doping knowledge questionnaire (*p* > 0.05). Students who engaged in sports demonstrated higher scores in anti-doping knowledge compared to those who did not participate in any type of exercise (56.10 ± 9.04 vs. 54.19 ± 9.10 points; *p* = 0.008). **Conclusions**: It was determined that the knowledge of essential anti-doping regulations and doping prevention strategies among bachelor’s degree medical students in Mexico was found to be suboptimal. Doping prevention should be included in the syllabus of the bachelor’s degree in medicine to help future doctors avoid professional errors, whether negligent or intentional.

## 1. Introduction

Athletic excellence has always been linked to pursuing new strategies to enhance performance, optimize recovery, and prevent injuries [1,2,3]. However, in recent years, the use of performance-enhancing substances has emerged as a significant concern in sports [4], as it poses serious ethical, health, and fairness issues [5,6]. For some, the use of performance-enhancing substances violates the principles of fair play and integrity that underpin sportsmanship. Additionally, some of these substances often have severe short-term and long-term health consequences, including cardiovascular issues, hormonal imbalances, and psychological disorders [7,8,9]. For this reason, the World Anti-Doping Agency (WADA) has established a clear code [10] distinguishing legal performance-enhancing substances, such as caffeine, nitrates, and creatine, from those classified as illegal (i.e., banned) due to their potential ergogenic effects associated with health risks for athletes, or violations of the spirit of sport. Amid this context, healthcare professionals, particularly medical practitioners, play a crucial role in safeguarding the integrity of sports by applying their expertise [11]. Physicians play a critical role in anti-doping efforts because they are often part of an athlete’s support team and are routinely responsible for prescribing medications to treat or prevent illnesses. Physicians possess extensive knowledge of pharmacology, yet during their university education, they are rarely taught which substances are prohibited in sports [12]. Additionally, physicians play a crucial role in identifying therapeutic alternatives that comply with anti-doping regulations. However, they are often not adequately trained in the fundamental principles of anti-doping rules during their formal education [5,12,13,14,15,16,17]. Finally, when an athlete temporarily requires a banned substance for legitimate medical reasons, physicians play a central role in applying for Therapeutic Use Exemptions (TUEs), ensuring compliance with anti-doping regulations [18]. Collectively, physicians are well-positioned to educate athletes about the risks of doping, including health complications, and to promote informed decision-making. However, for this role to be effective, they must receive adequate anti-doping education during their medical training—a practice that is not yet standard in most medical degree programs worldwide.

Physicians are classified as “athlete-support personnel” under The Code of the World Anti-Doping Agency (WADA), recognizing their critical role in the fight against doping [10]. According to The Code, ‘athlete-support personnel’ comprises any coach, trainer, manager, agent, team staff, official, medical, paramedical personnel, parent, or any other person working with, treating, or assisting an athlete participating in or preparing for sports competition. The Code also defines the role and competencies of physicians in relation to anti-doping in article 21.2: (a) to be knowledgeable of and comply with all anti-doping policies and rules adopted under The Code and which are applicable to them or the athletes whom they support; (b) to cooperate with the Athlete Testing Program; and (c) to use their influence on athlete values and behavior to foster anti-doping attitudes [10]. At the same time, the code defines the punishment of medical doctors in article 10.3.2. For violations of articles 2.7 (trafficking or attempted trafficking) or 2.8 (administration or attempted administration of prohibited substance or prohibited method), the period of ineligibility imposed shall be a minimum of 4 years up to lifetime ineligibility. The primary aim of The Code regarding athlete support personnel is to ensure that individuals involved in “physician-assisted doping” or aiding athletes in concealing doping practices face sanctions that are more severe than those imposed on athletes who test positive. The athlete is always responsible for any prohibited substance in his body (Section 2.1) under strict liability, but the period of ineligibility shall be reduced or even eliminated if an athlete can establish that they bear no fault or negligence [10]. Therefore, the World Anti-Doping Code places significant responsibility on physicians working with athletes, yet this responsibility is not typically supported by specific anti-doping education during medical school. Although WADA provides courses to educate physicians on anti-doping regulations, these are not mandatory and are likely less effective than incorporating such training into formal medical education.

Universities serve as crucibles for molding future healthcare providers, instilling not only medical expertise but also ethical frameworks crucial for their role as custodians of health. Yet, integrating anti-doping education within medical curricula remains an underexplored domain [5,17,19] Understanding the nuances of doping, its health implications, detection methods, and the ethical dimensions surrounding its use and prevention is imperative for medical students [11]. This knowledge equips them to address the needs of athletes but also to contribute to broader societal well-being by combating the misuse of performance-enhancing substances. Medical training in Mexico is provided, evaluated, and monitored by the Ministry of Public Education and Health. In Mexico, training in health sciences begins at the high school level, with the need to study chemical-biological sciences during high school as a pre-requirement to be admitted to a bachelor’s degree in medicine. Medical studies in Mexico are regulated mainly by the Mexican Official Standard [20], which establishes the relationships between the educational and health systems. The duration of the medical degree is generally 6 years, although each university creates its own academic program. In Mexico, no bachelor’s degree program in medicine currently includes a dedicated subject on anti-doping or the ethical considerations involved in treating athletes. Various studies on anti-doping knowledge among university students have been conducted globally [17,21,22,23,24,25,26], including in sports science, medicine, and pharmacy. Overall, these studies indicate suboptimal knowledge of anti-doping matters, even among students pursuing degrees linked to professions that support athletes. This gap is likely due to the absence of anti-doping-related subjects in their curricula. Two studies have demonstrated the effectiveness of anti-doping education programs specifically aimed at university medical students, positively influencing a variety of psycho-social factors, including attitudes [27,28] and confidence in discussing doping [28]. However, it is important to note that curricula and educational policies vary significantly across countries, and different educational systems may yield different outcomes. To date, there are no studies that have been conducted on anti-doping knowledge among medical bachelor’s degree students in Mexico, even though they play a crucial role in the fight against doping. Thus, the aim of this research was to evaluate the students’ understanding of anti-doping within the bachelor’s degree in medicine at Mexican universities. We hypothesized that medical students would have limited knowledge of basic anti-doping rules and strategies for ethically treating athletes. This hypothesis is based on the suboptimal anti-doping knowledge observed in other categories of athlete-support personnel, such as students pursuing a bachelor’s degree in sports sciences, who similarly lack formal anti-doping education during their training [22].

## 2. Materials and Methods

### 2.1. Participants and Study Design

The study sample was composed of 549 university students who completed a validated and reliable questionnaire about anti-doping knowledge between August 2022 and June 2024. All participants needed to be current students enrolled in a bachelor’s degree program in medicine from a Mexican university. They were selected from six universities belonging to the Anahuac Universities Network with analogous characteristics and contents (a 6-year program of 90 courses structured in 12 semesters and 500 academic credits). After concluding the 6-year program, an official certificate of studies legalized and apostilled by the Mexican Government is awarded to students who pass all program courses to be a certified physician.

A faculty member from each participating university administered the questionnaire during a class, providing an explanation of the study’s purpose and the questionnaire’s features. At this stage, the questionnaire was distributed via email exclusively to students enrolled in the class. It was presented online using Google Forms (Mountain View, CA, USA), ensuring that all students received the same set of questions in identical order with consistent response options. The use of an online form streamlined data collection while maintaining the anonymity of responses. Each class received a unique web link to the questionnaire, and once all students had completed it, the form was closed to prevent access by unauthorized individuals or unwanted participants.

Participation in the study was voluntary, with participants providing their consent by agreeing to a clause at the beginning of the questionnaire. Participants completed the questionnaire independently without referring to any external documents or websites, a process that was confirmed by the faculty member to replicate exam conditions. For each question, participants were required to select the correct answer from a list of options, without needing to input any information via keyboard. The faculty member addressed any inquiries unrelated to the anti-doping content of the questionnaire. After completing the questionnaire, participants received information about their scores but did not learn which answers were correct or incorrect.

Thirty-one students were removed from the study due to their failure to answer all questions in the questionnaire. The procedures developed in this investigation were approved by an institutional Ethics Review Committee of the Francisco Vitoria University (IRB number UFV 26/2022) and the study was carried out in accordance with the procedures approved by the Declaration of Helsinki. Specific information about the study sample can be found in Table 1.

### 2.2. Questionnaire

This study followed a similar pattern to a previous publication in which the anti-doping knowledge of bachelor students enrolled in sports sciences programs at universities in Spain [22]. The questionnaire employed in this study was an adapted version of the “WADA Play True Quiz” [29]. The adapted version was used instead of the original version for three reasons: (a) in the original version, participants are addressed as athletes; (b) there are ten questions that inquire about behaviors/attitudes toward doping rather than anti-doping knowledge; and (c) there are no questions associated with sociodemographic variables. This modified questionnaire included 36 true/false and multiple-choice questions focused on fundamental anti-doping knowledge. The questionnaire used in this study was the same as that employed by Aguilar-Navarro et al. [22] in their research with sports science university students. Readers can refer to that study for the full questionnaire, which includes all 36 questions, potential answers, and the correct responses, available as open access in the appendix. Each question assessing anti-doping knowledge was worth one point, with a maximum possible score of 36. This score was then converted to a 0–100-point scale for better clarity regarding the participants’ overall level of anti-doping knowledge. Good reliability of the questionnaire (coefficient of variation of 3.6% and an intraclass correlation coefficient of 0.77) has been previously reported when re-testing the overall anti-doping knowledge in 56 university students after 4 weeks [22]. Additionally, the questionnaire collected socio-demographic information about participants, including their gender, university, degree program, involvement in sports, membership in a sports federation, and whether they were currently using dietary supplements.

### 2.3. Statistical Analysis

Following data collection, the information was organized, verified, and analyzed using the SPSS v27.0 statistical package. The participant counts in the various sociodemographic categories were expressed by frequencies and percentages. Data normality for the questionnaire scores was assessed using the Kolmogorov-Smirnov test, the Shapiro–Wilk test, and histograms. The score obtained in the questionnaire was normally distributed overall and, in the subgroups, analyzed, and it was expressed in mean ± standard deviation. Differences in scores obtained among groups were analyzed using unpaired *t*-tests (two categories) or by using a one-way analysis of variance (ANOVA; three or more categories). The significance level was set at *p* < 0.050.

## 3. Results

From the total sample, 63.9% of the students were female, 34.8% were male, and 1.3% with non-binary sex. A total of 23.7% of students enrolled in the first course of the degree, 24.8% in the second course, 18.2% in the third course, 17.7% in the fourth course, 14.6% in the fifth and only 1.1% in the sixth course. Only 77 (14%) students had received anti-doping information/lessons in any of the subjects of the degree. Within the sample, 328 students (59.7%) engaged in some form of exercise or sport at least three times a week, while the remaining 40.3% did not meet this level of activity. Among those who participated in sports, a total of 23 different sports modalities were represented. A total of 81.1% of the students played team sports, and the remaining 18.9% of the sample played individual sports. Twenty students (3.6%) were registered in a national sports federation at the time of the questionnaire filling. Seventy-four students (13.5%) indicated that they were using dietary supplements when they completed the questionnaire (Table 1).

On average, the students obtained a score of 55.30 ± 9.08 points (range = 28–83 points). From the total, 23.0% of students obtained a score below 50 points and 73.4% of students obtained a score between 50 and 69 points. Only 3.6% of the respondents obtained a score above 70 points (Figure 1).

Question 15, related to the definition of the acronym TUE, which stands for Therapeutic Use Exemption, had the lowest error rate at 12.2% (Figure 2). Question number 30, associated with who determines whether the submission of a therapeutic use authorization is approved or rejected, had the highest error rate (86.5%). Overall, there were 14 questions where the error rate exceeded 50% among the participants.

Males and females had similar scores (55.48 ± 8.96 vs. 55.20 ± 9.16 points, respectively; *p* = 0.442). There was no effect of the course on the score obtained in the anti-doping knowledge questionnaire (*p* > 0.05; Figure 3).

The scores of students who reported receiving anti-doping information or lessons were slightly lower than those of students who had not received any anti-doping instruction during their degree (54.07 ± 8.04 and 55.50 ± 9.24 points, respectively; *p* = 0.101). Students affiliated with sports federations scored higher in anti-doping knowledge compared to those who were not affiliated with any sports federation (58.89 ± 7.24 vs. 55.17 ± 9.12 points, respectively; *p* = 0.036; Figure 4).

Students who used supplements had comparable anti-doping knowledge scores to those who did not use dietary supplements (55.22 ± 8.58 vs. 55.32 ± 9.17 points, respectively; *p* = 0.465). The students who participated in any form of exercise or sport had a higher anti-doping knowledge score compared to those who did not engage in physical activity (56.10 ± 9.04 vs. 54.19 ± 9.10 points, respectively; *p* = 0.008; Figure 5). Students involved in individual sports scored higher on the anti-doping knowledge questionnaire than those participating in team sports (56.34 ± 9.00 vs. 54.79 ± 9.20 points, respectively; *p* = 0.111).

## 4. Discussion

The purpose of the present study was to evaluate the knowledge of anti-doping among students enrolled in the bachelor’s degree program in medicine at various universities in Mexico. This investigation was motivated by the observation that anti-doping education is absent from the undergraduate medical curriculum despite its purpose of preparing future physicians who may play critical roles in working with athletes across various performance levels. To achieve this objective, we employed a modified version of the WADA Play True Quiz, a survey designed to evaluate fundamental anti-doping knowledge among athletes and support personnel. The modifications to the Play True Quiz were limited to rephrasing questions originally framed to be answered from an athlete’s perspective. In these instances, the wording was adjusted to reflect the perspective of “a member of the athlete support personnel,” clearly guiding students to respond as if they were future physicians working with athletes. With this perspective, we collected data on anti-doping knowledge from students at six distinct Mexican universities that shared a similar curriculum for the bachelor’s degree in medicine. The findings of this study reveal a concerning level of anti-doping knowledge among these medical students, as evidenced by a mean score of 55.30 ± 9.08 points (range = 28 to 83 points) on a questionnaire designed to assess essential anti-doping knowledge (Figure 1). These results are consistent with previous research that has shown that sports physicians and general medicine practitioners lack the required knowledge on the prohibited list and methods [5,12,13,14,15,16]. Overall, these data highlight a significant gap in the educational training of medical students on the critical issue of combating doping in sports. Interestingly, students who were enrolled in a sports federation, exercised at least three days per week, or engaged in an individual sport demonstrated significantly higher anti-doping knowledge. The difference in knowledge between students who participate in sports and those who do not could be explained by greater prior exposure to anti-doping information. Students who participate in sports are more likely to be exposed to anti-doping information through coaches, seminars, and athlete-specific educational materials. Furthermore, anti-doping organizations often focus on educating athletes and their entourage about prohibited substances and the consequences of doping. This suggests that these students may be gaining anti-doping awareness through informal education channels outside the university setting. Collectively, this information underscores the need to incorporate anti-doping education into the curriculum for future Mexican physicians. This could be achieved through a mandatory course for all medical students or an optional course tailored for those expressing an interest in pursuing sports medicine during their university training. Interestingly, several studies have specifically shown the effectiveness of anti-doping education programs for university students [26,27,28,30]. Specifically, these studies indicated that ad hoc educational programs enhanced students’ knowledge of anti-doping, decreased pro-doping attitudes [27,28], and positively influenced their moral perspectives towards doping behaviors [26]. Collectively, these studies demonstrate that anti-doping education programs can improve knowledge and awareness of doping-related issues among university students, with tailored and structured educational interventions proving efficacy in enhancing understanding, shaping attitudes, and promoting compliance with anti-doping regulations.

According to data from the National Laboratory of Prevention and Doping Control of Mexico, the incidence of adverse analytical findings in doping controls of Mexican athletes was 3.8% in the 2009–2015 interval [31]. The proportion of adverse analytical findings in Mexico is slightly higher than the prevalence of doping worldwide (~1.9% [32]). Doping is a sociological phenomenon deeply influenced by cultural, societal, and economic factors, which helps explain the significant variations in doping prevalence observed across different countries. For example, cultural attitudes toward competition, societal pressure to achieve athletic success, and the availability of resources for anti-doping efforts all play a role in shaping doping behaviors among athletes [33]. These data highlight the need to raise awareness among health professionals regarding doping in Mexico and the urgency of including anti-doping information in the syllabus of the bachelor’s degree in medicine. Interestingly, data on adverse analytical findings in Mexico indicates that the sports with the highest incidence are baseball, cycling, and athletics [31]. Therefore, Mexican physicians treating athletes in these sports should be particularly vigilant about the increased likelihood of their athletes engaging in doping practices. Physicians in Mexico should receive specialized training to support athletes in adhering to anti-doping regulations. This includes utilizing their medical expertise and pharmacological knowledge to ensure prescribed medications comply with anti-doping rules, thereby preventing unintentional violations, and educating athletes on the health risks and broader implications of doping.

The National Mexican Anti-Doping Organization is the body in charge of guaranteeing fair play and doping-free play in Mexican athletes through the implementation of different anti-doping programs [34], including education, research, and policy initiatives. This national anti-doping organization promulgates programs harmonized with the guidelines and directives proposed by the latest version of The Code [10] and with the International Standard for Education [35]. The National Mexican Anti-Doping Organization has recently promoted several educational and learning initiatives for students undertaking a bachelor’s degree in medicine. However, these seminars are offered in a limited number of Mexican universities and do not reach a wide proportion of students. Specifically, since 2018, the National Mexican Anti-Doping Organization has had a documented education program, which includes all educational modalities: face-to-face workshops and seminars, education at events, online learning, promotion, and delivery of information through its website and social networks. This educational plan aims to provide Mexican athletes and their support staff with valuable anti-doping information in their amateur and/or professional careers. However, this educational plan is not mandatory and reaches a limited number of personnel working with athletes. This educational initiative is likely insufficient to ensure that all Mexican physicians working with athletes receive adequate anti-doping training, particularly in a country with 128 million people.

Due to the geographical characteristics of Mexico (a land area of approximately 1.96 million square kilometers) and its population, the National Mexican Anti-Doping Organization has focused most of its educational efforts on the use of the ADEL Online Anti-Doping Training Platform created by WADA [36]. ADEL offers access to all topics related to clean sport attitudes and the fight against doping. It offers courses for athletes, coaches, doctors, administrators, and all the people interested in learning more about the fight against doping and the protection of the values of fair play. Since 2022, authorities, coaches, technicians, doctors, and athletes participating in national and international sporting events have been required to take the ADEL online courses proposed by the National Mexican Anti-Doping Organization. However, this educational initiative is mainly aimed at international-level athletes with less influence on amateur athletes and athlete-support personnel so far. As the National Mexican Anti-Doping Organization in its 2022 education plan [37], its greatest challenge for the coming periods is to have a more significant impact on medical professionals since only 9% of the participants in the ADEL program in Mexico are physicians. This highlights that such “open” courses are ineffective, as they fail to reach medical practitioners adequately. As previously discussed, this underscores the need for anti-doping education to be incorporated into the formal medical training of Mexican physicians.

The current findings align with these previous investigations carried out with health professionals from other countries such as Serbia, Slovenia, Italy, France, the United Kingdom, Greece, Ireland, and India [5,12,13,14,15,16]. However, comparing the level of anti-doping knowledge in this study with other investigations is challenging due to the use of different questionnaires to assess anti-doping knowledge across studies. Even without the possibility of direct data comparisons, the findings of these studies consistently highlight that inadequate anti-doping knowledge among healthcare professionals is a global issue. In the current study, the average score achieved by the students was 55.30 ± 9.08 points out of a possible maximum score of 100 points. Additionally, only 3.6% of participants scored above 70 points (Figure 1). These findings are troubling, especially considering that the modified version of the WADA Play True Quiz used in this study consisted solely of fundamental and straightforward questions related to anti-doping regulations and prevention strategies. Interestingly, students pursuing a bachelor’s degree in sports science in Spain were tested using the same adapted version of the WADA Play True Quiz. Sports science students in Spain demonstrated significantly higher rates of correct responses, with a mean score of 65.78 ± 10.11 [22]. Interestingly, as in Spain, Mexican students who engaged in various forms of exercise or were affiliated with a sports federation showed higher anti-doping knowledge scores (Figure 4 and Figure 5). This suggests that a significant portion of students’ anti-doping knowledge may be derived from sources external to their formal academic training.

In total, there were fourteen questions out of the thirty-six that composed the questionnaire, where the rate of error was superior to 50% (Figure 2). Despite a high percentage of respondents (87.8%) knowing the meaning of the acronym TUE (Therapeutic Use Exemption), most of them (86.5%) did not know who determines whether the therapeutic use authorization is approved or rejected. This shows that students lack knowledge about the procedures related to TUEs, a key function of physicians working with athletes. Only 18.8% of respondents correctly distinguished between a banned substance and an ergogenic aid (i.e., a legal performance-enhancing substance), and only 44.6% acknowledged that the WADA list of prohibited substances and methods is applicable across all sports. A significant number of students understood that refusing to undergo doping control can result in penalties comparable to those for a positive test (77.9%). Conversely, 72.1% of respondents incorrectly believed that an athlete has the right to refuse entry to a Doping Control Officer conducting an out-of-competition test to protect their privacy. Approximately 71.2% of respondents were not aware that holding a prohibited substance or method constitutes a violation of anti-doping rules, which carries penalties similar to those for other breaches, like the detection of a banned substance in urine or blood tests. For this reason, it is crucial that athlete support personnel, including physicians, are familiar with and fully understand Article 21 of The Code (“Additional Roles and Responsibilities of Athletes and Other Persons”). This is especially true for Section 2, which outlines the specific responsibilities of athlete support personnel in detail [10].

This investigation presents some limitations that should be discussed to improve the results’ applicability. To date, there are no worldwide standardized research methods to assess the anti-doping knowledge of athlete support personnel [38]. This limits the interpretation and comparison of the results with previous studies, as mentioned above. However, we used a validated and reliable questionnaire previously used in Spain to assess anti-doping knowledge in sports science university students [22], thus allowing comparison between different members of athlete support personnel in different countries. Although all medical students from different Mexican universities were invited to participate in this research, lower rates of responses were obtained from 6th-year students who do not regularly attend classes on the university campus due to their involvement in internships and community service. Additionally, the response rate was relatively low in male students, of the total sample, only 34.8% were male. Another limitation related to the study sample is that all participants were students from universities within the Anahuac Universities Network. Although the medical degree program in this network shares similar characteristics and content with other medical programs in Mexico, future research should include students from different universities to ensure broader representation. Similarly, the findings of this study cannot be generalized to students in other healthcare-related programs, such as nutrition or physiotherapy, who may also become future healthcare professionals and athlete support personnel. Assessing these groups in future studies, both in Mexico and internationally, would provide a more comprehensive understanding of anti-doping knowledge across different disciplines. Despite these limitations, the authors believe that the article presents valuable information for the university community and anti-doping regarding university students’ lack of anti-doping knowledge as future members of athlete support personnel in Mexico.

## 5. Conclusions

The understanding of fundamental anti-doping regulations and strategies for preventing doping among undergraduate students in medicine at various universities in Mexico was suboptimal. This is because the adapted version of the WADA Play True Quiz featured basic questions regarding anti-doping. There is a need to include doping prevention education programs in the syllabus of the bachelor’s degree in medicine to help future Mexican physicians avoid professional errors, whether negligent or intentional [5,17]. We advocate for the inclusion of a mandatory course for all medical students, or an optional course tailored for those expressing an interest in pursuing sports medicine during their university training. It would be desirable for physicians who regularly work with athletes to be familiar with the prescription regulations about anti-doping [11], not only because it may lead to a sanction for the physician but also because the use of a banned substance may cause a sanction for the athlete who receives the prescription along with potential medical consequences [5].

## Figures and Tables

**Figure 1 healthcare-13-00742-f001:**
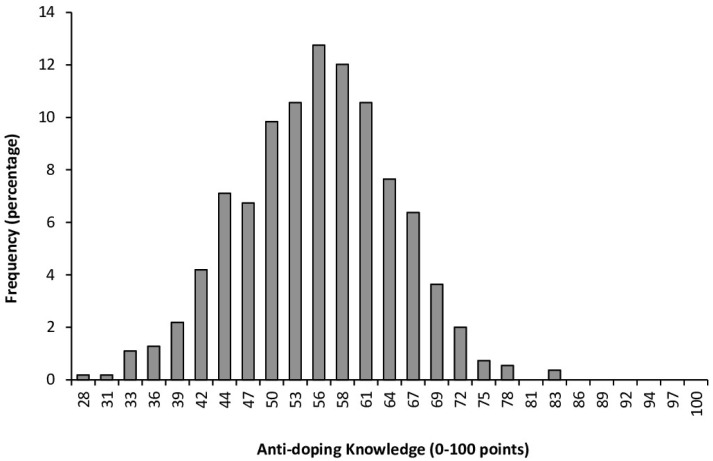
Frequency of Mexican bachelor students studying a degree in medicine based on their understanding of anti-doping as evaluated by a modified version of the WADA Play True Quiz.

**Figure 2 healthcare-13-00742-f002:**
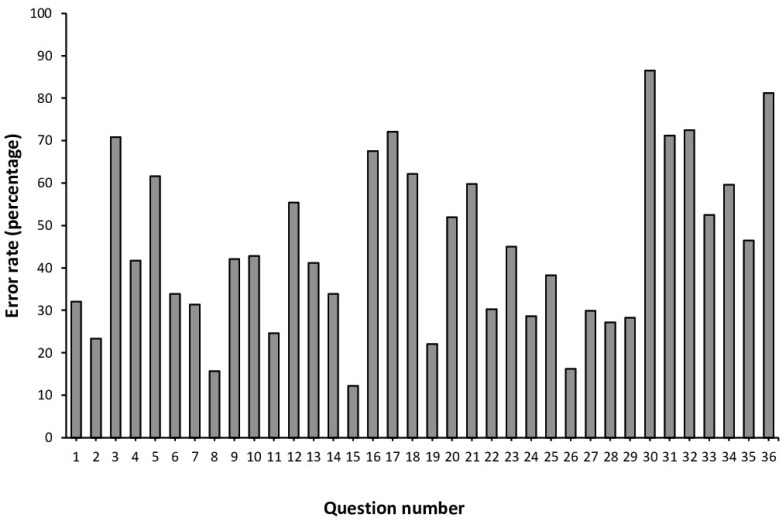
Rate of error in the questions of the adapted version of the WADA Play True Quiz in Mexican bachelor students of the degree in medicine.

**Figure 3 healthcare-13-00742-f003:**
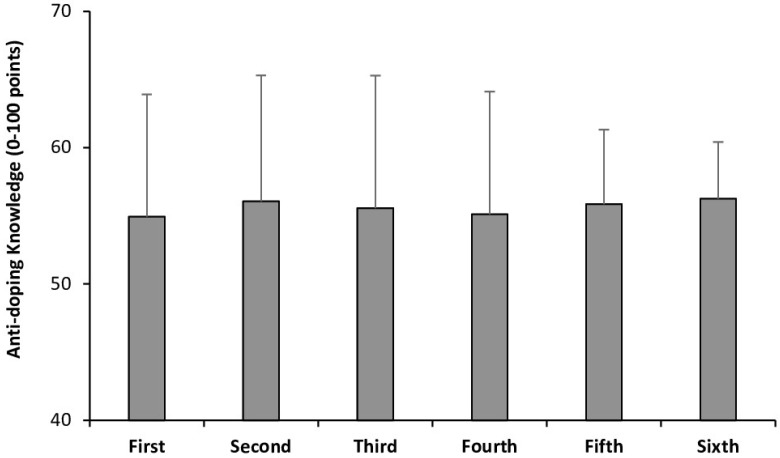
Scores in the adapted version of the WADA Play True Quiz of students undertaking a degree in medicine in Mexican universities according to their course. Data for each course are presented as mean ± standard deviation.

**Figure 4 healthcare-13-00742-f004:**
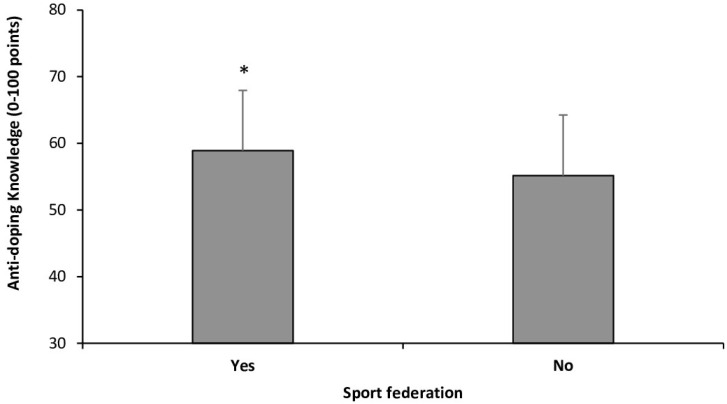
Score in the adapted version of the WADA Play True Quiz in Mexican bachelor students registered in sports federations. (*) Different from students not registered in sports federations at *p* < 0.050.

**Figure 5 healthcare-13-00742-f005:**
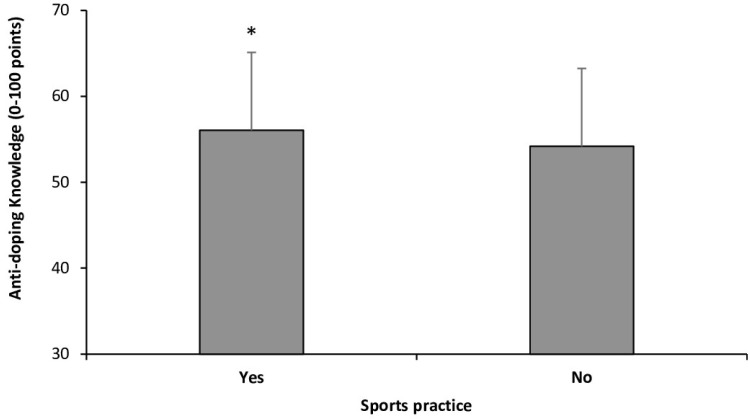
Score in the adapted version of the WADA Play True Quiz in Mexican bachelor students who practiced any form of exercise or sport. (*) Different from students who did not practice any form of exercise at *p* < 0.050.

**Table 1 healthcare-13-00742-t001:** Anti-doping knowledge scores by sociodemographic variables among bachelor students undertaking a degree in medicine at Mexican universities.

	Variable	N (Frequency)	Score (Points)	*p*-Value
Gender	Male	191 (34.8%)	55.48 ± 8.96	0.391
	Female	351(63.9%)	55.20 ± 9.16	
	Non-binary	7 (1.3%)	47.22 ± 15.21	
	Total	549 (100%)		
Course	First	130 (23.7%)	54.92 ± 8.99	0.877
	Second	136 (24.8%)	56.06 ± 9.24	
	Third	100 (18.2%)	55.56 ± 9.72	
	Fourth	97 (17.7%)	55.12 ± 9.01	
	Fifth	80 (14.6%)	55.86 ± 5.46	
	Sixth	6 (1.1%)	56.25 ± 4.17	
Anti-doping lessons	No	472 (86%)	55.50 ± 9.24	0.101
	Yes	77 (14%)	54.07 ± 8.04	
Sport practiced	No	221 (40.3%)	54.19 ± 9.10	0.008
	Yes	328 (59.7%)	56.10 ± 9.04	
Type of sport	Individual	62 (18.9%)	56.34 ± 9.00	0.111
	Team-based	266 (81.1%)	54.79 ± 9.20	
Registered in a sport federation	No	529 (86.5%)	55.17 ± 9.12	0.036
	Yes	20 (3.6%)	58.89 ± 7.24	
Use of dietary supplements	No	475 (86.5%)	55.32 ± 9.17	0.465
	Yes	74 (13.5%)	55.22 ± 8.58	

Note. Values are presented as number (frequency) for categorical variables and as mean ± standard deviation for continuous variables. The *p*-value corresponds to the statistical comparison of anti-doping knowledge scores between groups using an independent *t*-test (for two categories) or one-way ANOVA (more categories). A *p*-value < 0.05 indicates a statistically significant difference between groups.

## Data Availability

The original contributions presented in this study are included in the article. Further inquiries can be directed to the corresponding author.

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
