# Peer review of "Anti-Doping Knowledge Among Medical Bachelor’s Degree Students in Mexico"

_healthcare, 2025, doi:10.3390/healthcare13070742_

Round 1
Reviewer 1 Report
Comments and Suggestions for Authors
Dear Author
After reviewing your manuscript, I conclude that this manuscript needs improvement, this is important so that your manuscript is much better quality and worthy of being published in this extraordinary journal.
Thank you

Author Response
"Please see the attachment."

Reviewer 2 Report
Comments and Suggestions for Authors
The introduction is well thought out. However, it would be interesting to learn more about anti-doping training and its impact on medical practice.
The methodology is correct, but I would like to know the choice of the WADA questionnaire, i.e. why this questionnaire. Also, the adaptation of the WADA questionnaire for support staff is correct, but was this adaptation specifically validated for medical students?
The difference in knowledge between students who are involved in sports and those who are not is interesting. Could it be explained whether this difference is related to a greater prior exposure to anti-doping information?
When mentioning the lack of specific anti-doping training in medical education, it would be useful to propose how to integrate this training, i.e. through the provision of specific courses, seminars, lectures?
Why were only universities from the Anahuac Network selected, and was it considered whether this sample is representative of all medical universities in Mexico?
The majority of students obtained average scores, but is there a defined threshold that indicates an acceptable level of anti-doping knowledge? This would help to better interpret the results.
Overall the study is very good, but check those small comments that in my view would help to make the study clearer.
Author Response
"Please see the attachment."

Round 2
Reviewer 1 Report
Comments and Suggestions for Authors
Dear author
After I observed, the manuscript can be improved and enhanced well. Thus my conclusion for this manuscript is accepted for publication.